# Kernel Identification Through Transformers

**Fergus Simpson**
Secondmind
Cambridge, UK
`fergus@secondmind.ai`

**Ian Davies** *
InstaDeep
London, UK

**Vidhi Lalchand**
University of Cambridge
Cambridge, UK

**Alessandro Vullo**
Secondmind
Cambridge, UK

**Nicolas Durrande**
Secondmind
Cambridge, UK

**Carl Rasmussen**
University of Cambridge
Cambridge, UK

## Abstract

Kernel selection plays a central role in determining the performance of Gaussian Process (GP) models, as the chosen kernel determines both the inductive biases and prior support of functions under the GP prior. This work addresses the challenge of constructing custom kernel functions for high-dimensional GP regression models. Drawing inspiration from recent progress in deep learning, we introduce a novel approach named *KITT: Kernel Identification Through Transformers*. KITT exploits a transformer-based architecture to generate kernel recommendations in under 0.1 seconds, which is several orders of magnitude faster than conventional kernel search algorithms. We train our model using synthetic data generated from priors over a vocabulary of known kernels. By exploiting the nature of the self-attention mechanism, KITT is able to process datasets with inputs of arbitrary dimension. We demonstrate that kernels chosen by KITT yield strong performance over a diverse collection of regression benchmarks.

## 1 Introduction

In recent years deep parametric models have become a prominent class of model for supervised learning and have delivered impressive empirical performance over a wide range of tasks. An important limitation, however, is that in their conventional form deep models do not provide prediction uncertainty. While their Bayesian counterparts try to achieve this, they require significant modifications to the training procedure and are computationally expensive. Uncertainty quantification in deep models is widely considered to be an open problem, the large array of research proposing alternative Bayesian neural networks underscores this [10, 12, 17].

On the other hand, kernel driven methods within the Bayesian framework like Gaussian processes (GPs) account for prediction uncertainty by design. While GPs provide a flexible framework for inferring distributions over functions, the inductive biases are controlled by the kernel function[1]. A well chosen kernel will typically yield dramatically better performance than a poorly chosen one.

How should we learn expressive kernels for high-dimensional tasks? This has frequently been highlighted as a central question for the continued relevance of GP methods [11]. This work uses representations generated by a deep neural network to identify suitably expressive kernels for high-dimensional GP regression tasks. Kernel recommendation is performed by a decoder with access to a large vocabulary of primitive kernels and products of primitive kernels. The decoder maps an

---

*Work undertaken while at Secondmind

[1]also called covariance function or covariance kernel

35th Conference on Neural Information Processing Systems (NeurIPS 2021).

encoded representation of a dataset to a kernel that can be used to model it. The representation is attained by encoding the dataset, treated as a sequence of (input, output) pairs $\mathcal{D} = \{\boldsymbol{x}_i, y_i\}_{i=1}^N$, utilising the permutation-equivariant nature of self-attention networks.

By training KITT with a sufficiently rich vocabulary of kernels, it can predict suitable kernels for a diverse array of real datasets. This work presents the following novel contributions:

- Inspired by the successes of image captioning networks, we develop a novel framework named KITT for amortised kernel search. KITT takes raw datasets for predictive modelling as input and proposes kernels composed from a large vocabulary of kernel functions.
- KITT's architecture introduces two novel features: it is entirely agnostic to the length and dimensionality of the data we wish to perform inference on, and it offers double permutation invariance (this ensures its outputs are invariant to permutations in either input dimensions or data points).
- We show that KITT can deliver kernel predictions in under 0.1 seconds.
- We introduce a novel variant of the linear kernel which forms a key component of KITT's vocabulary.
- We demonstrate that the kernels identified by KITT offer strong performance against other baselines which deal with kernel engineering in the context of GPs.

## 2 Background

This section offers a brief review of the two topics which are central to this work, namely Gaussian Processes and Transformers.

**Gaussian Processes.** GPs offer a highly versatile framework for predictive modelling [23] with generalisation properties controlled by a kernel function parameterised by hyperparameters. The functional form of the kernel governs the global attributes of the supported functions, such as smoothness and periodicity. However, a suitable kernel function is unknown *a priori* and the choice of the kernel function is a fundamental model selection problem. Once a kernel has been chosen, training conventionally proceeds by learning a point estimate of the hyperparameters that maximise the GP log marginal likelihood $\boldsymbol{\theta}^* = \operatorname{argmax}_{\boldsymbol{\theta}} \log p(\boldsymbol{y}|\boldsymbol{\theta})$. The marginal likelihood is available in closed-form for models with Gaussian likelihoods. Below we briefly summarise the standard GP framework.

GPs are distributions over functions from which one can sample realised function values for given inputs. Concretely, for observations $X = \{\boldsymbol{x}_i\}_{i=1}^N$, and positive definite kernel function $k_\theta(\cdot, \cdot)$, with hyperparameters $\boldsymbol{\theta}$, $f(\cdot) \sim \mathcal{GP}(\mathbf{0}, k_\theta)$. Typically, we observe noisy realisations of the latent function which are corrupted with Gaussian noise, $y_i = f(\boldsymbol{x}_i) + \epsilon_i$, $\epsilon_i \sim \mathcal{N}(0, \sigma_n^2)$, and infer the kernel hyperparameters through maximising the likelihood of the model.

The GP marginal likelihood objective, $p(\boldsymbol{y}|\boldsymbol{\theta})$, is obtained by marginalising the likelihood $\boldsymbol{y}|\boldsymbol{f} \sim \mathcal{N}(\boldsymbol{f}, \sigma_n^2 \mathbb{I})$ over the prior $\boldsymbol{f}|X, \boldsymbol{\theta} \sim \mathcal{N}(\mathbf{0}, K_\theta)$,

$$p(\boldsymbol{y}|\boldsymbol{\theta}) = \int p(\boldsymbol{y}|\boldsymbol{f})p(\boldsymbol{f}|\boldsymbol{\theta}) \, d\boldsymbol{f} = \mathcal{N}\left(\mathbf{0}, K_{\boldsymbol{\theta}} + \sigma_n^2 \mathbb{I}\right), \tag{1}$$

where $\boldsymbol{f} = f(X)$ denotes a vector of realised function values, and $K_\theta$ denotes the $N \times N$ covariance matrix corresponding to evaluations of the covariance function at the $N$ training inputs, $(K_\theta)_{i,j} = k_\theta(\boldsymbol{x}_i, \boldsymbol{x}_j)$.

A long standing question is how best to select an appropriate kernel function for a given task. One approach is to search over a discrete space of kernels, defined by combining a selection of primitive kernels with a predefined grammar [2, 6]. Typically, a greedy search is performed to identify the kernel offering the best representation of the data. Ideally, the quality of the kernel is quantified by the Bayesian model evidence, which can be computed by marginalising the marginal likelihood over the hyperparameters. However, since the integral is challenging to compute, each kernel's suitability is instead usually determined via a proxy for the model evidence, such as the Bayesian Information Criterion (BIC) [25].

For a principled, Bayesian approach to kernel design, instead of selecting a single kernel, one ought to consider multiple candidates. In other words, it is desirable to marginalise over the space of

kernels, not only over the space of functions for a single kernel.

$$p(\boldsymbol{y}|\mathcal{D}) = \sum_i p(\boldsymbol{y}|K_i, \mathcal{D})p(\mathcal{D}|K_i)p(K_i)\,. \tag{2}$$

This yields a rich posterior distribution comprised of a mixture of Gaussians. A conventional kernel search makes three key approximations. First, that contributions from all but the single chosen kernel $K^*$ can be neglected; second, that contributions from all but the maximum likelihood hyperparameters $\theta^*$ can be neglected; third, that the proxy for the model evidence is a reliable one. There are several regimes, for example where the data is sparse or noisy, where all three of these assumptions do not hold. We shall aim to improve upon all three of these issues.

**Transformers**  Transformers are a form of deep neural network which rely upon the attention mechanism [32] to capture global context. While they were originally proposed to tackle machine translation tasks [29], they have rapidly attained state-of-the-art performance in a number of other areas of machine learning [14, 21, 22]. Of particular relevance to this work, they have been successfully applied to image captioning [5, 33], a task which involves summarising the key characteristics of rich data in a grammatical form. This has a striking parallel to the challenge of selecting an appropriate kernel, especially since a form of grammar can be used to construct a broad selection of GP kernels.

The self-attention mechanism of transformers naturally lends itself to the permutation-invariant setting, as demonstrated by Zaheer et al. [34]. This invariance to permutations in the ordering of inputs has been exploited to create Set Transformers, introduced in Lee et al. [18]. The AHGP model [19] uses a variant of the Set Transformer to infer the hyperparameters of the spectral mixture kernel.

Like all deep networks, transformers thrive when presented with an abundance of training data. Fortunately, for the task we have at hand, the training set is unlimited in size as we may sample training data from GPs with known kernels.

As stressed by Liu et al. [19], selecting an architecture which reflects the appropriate invariances is vital. The AHGP model is designed to be invariant to permutations in the ordering of the datapoints. We go one step further, and introduce a model which is doubly permutation invariant: its output is also invariant to permutations in the input dimension (whereas AHGP is *equivariant* to the input dimensions).

## 3  KITT

In this section we describe and motivate KITT, a network which takes as inputs a set of datapoints $\{\boldsymbol{x_i}, y_i\}$, and outputs a kernel recommendation in the form of a 'caption', by utilising a kernel grammar. The code is available at `https://github.com/frgsimpson/kitt`.

**Kernel Grammar and Vocabulary:**  Throughout the GP literature, the most commonly used kernels belong to a limited set of primitive functions. In this work, we utilise eight primitive kernels: the squared exponential; periodic; white noise; three variants of the Matern kernel $\left(\frac{1}{2}, \frac{3}{2}, \frac{5}{2}\right)$; the cosine kernel, and our novel variant of the linear kernel. This list comprises six stationary isotropic kernels, one stationary anisotropic kernel (cosine), and one non-stationary kernel (linear).

From this small set of primitive kernels, we wish to construct a larger array of more expressive kernels. This can be achieved by leveraging the closure properties of kernel functions [27]. Permitted operations include addition, multiplication, convolution, composition, and affine transformations. For simplicity, this work shall only consider two operators: addition and multiplication.

Whilst this grammar of addition and multiplication appears superficially simple, it has some idiosyncrasies that would make it challenging for a network to learn. For example, if a network is unaware that multiplication is commutative, it would have to learn to recognise $k_1 * k_2$ and $k_2 * k_1$ separately, for all combinations of $k_1$ and $k_2$. It would also need to learn that multiplying a noise kernel with another stationary kernel yields another noise kernel. Encoding this information *a priori* greatly facilitates the learning process. To achieve this, we enlarge the vocabulary by defining product kernels as single 'words', rather than incorporating products as part of the grammar (the full list is provided in the Appendix). This allows us to exclude redundant combinations from consideration. For example, only a single token is used to represent both $k_2 * k_1$ and $k_1 * k_2$. While this enlarges

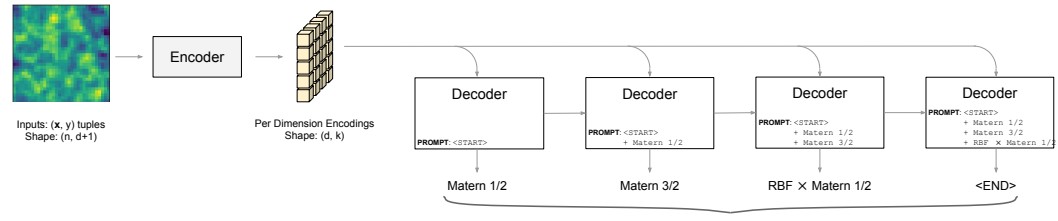

Figure 1: The architecture for KITT, partly motivated by image captioning networks, which also act to transform a rich dataset into a grammatical expression of its contents.

the vocabulary, we can be confident that the network will be able to cope, since there will still be far fewer 'words' than can be found in natural language tasks where transformers are known to excel. We use products of two primitive kernels as part of the base vocabulary and found that incorporating higher order products do not significantly change performance.

A further advantage of defining product kernels at the vocabulary level is that we need no longer include operators inside the vocabulary. The multiplication is already baked into the expanded set of kernels, while the addition takes place implicitly, much like the white space between words in a natural language task. This precludes the construction of nested structures of operators, which would allow for even richer kernels. However, since this work represents a first attempt at performing a kernel search with a neural network, we choose to keep the captioning task to be a relatively simple one, and leave more complex grammatical compositions as an opportunity for future exploration. Even with this relatively simple grammar, if we permit a caption of four *words* from a base vocabulary of 32, we are effectively searching across a space of around $36\,000$ kernels.

**Priors.** While the priors we impose upon the hyperparameters will have some impact during optimisation, it is their influence upon the generation of KITT's training data that is of central importance to this work. Random samples are drawn from the hyperparameter priors $p(\theta)$ and input locations $x \sim U(-2.5, 2.5)$ before each random sample of $y$ is generated. The variances and lengthscale parameters of all kernels (including product kernels) are assigned lognormal priors, such that $\log \theta \sim \mathcal{N}(0, 1)$.

The cosine kernel is unique among KITT's vocabulary, in that it is inherently anisotropic, which can be important if a preferred direction exists within the data. Samples drawn from the kernel manifest as plane waves which propagate along a characteristic direction of the kernel. If we were to impose a lognormal prior (or any other positively constrained prior) on its lengthscales, this would restrict the direction of the kernel to a small fraction $2^{1-D}$ of its permitted parameter space. We therefore adopt a different approach in this case, assigning $\ell \sim \mathrm{Cauchy}(0, 5)$ for the lengthscales.

**Training.** KITT is trained entirely on synthetic data. Each training example is generated from a randomly selected kernel, with a randomly drawn set of hyperparameters. This data could be generated on the fly, but since this can be a computationally expensive process, we generated a training set in advance which comprised of $200,000$ labeled examples. While the model can be constructed for an arbitrary number of inputs points and input dimensions, during training we restrict ourselves to the case of 4 input dimensions and 64 input points per sample. The loss function corresponds to $-\log p(k|\mathcal{D})$, the negative log probability the network assigned to the correct term in the vocabulary. The Adam optimiser was used with an initial learning rate of $10^{-4}$, and a decay schedule with a decay rate of $0.1$ every $50,000$ iterations. Due to the relatively noisy nature of the classification task, a relatively large batch size of 128 was found to be beneficial. The vocabulary included product kernels of at most two terms in addition to the primitive kernels, yielding a final vocabulary of size 34.

**Heteroscedastic noise.** When taking the product between the white noise and any stationary kernel, we recover another noise kernel. These redundant expressions are omitted from KITT's vocabulary. However, the result of the product between the noise kernel and the (non-stationary) linear kernel merits special attention. The linear kernel is defined as

$$K_{\mathrm{LIN}}(\boldsymbol{x}, \boldsymbol{x'}) = \boldsymbol{\sigma}^2 (\boldsymbol{x} - \boldsymbol{c})(\boldsymbol{x'} - \boldsymbol{c}), \tag{3}$$

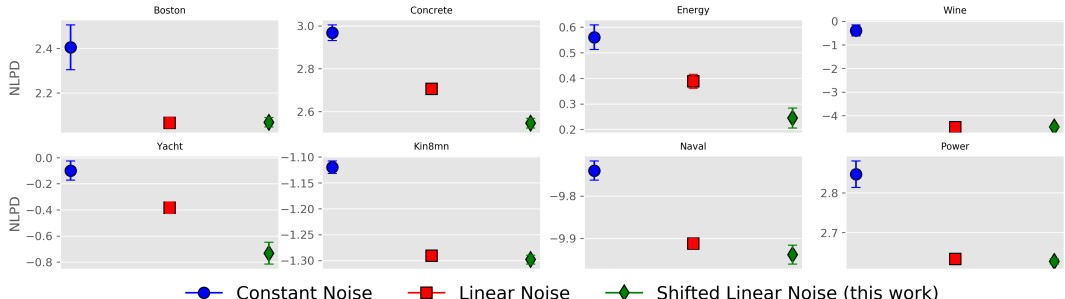

Figure 2: Negative log likelihood values for three different noise models when used alongside the RBF kernel. All of the datasets clearly benefit from the modelling of heterescedastic noise, while three benefit from the additional freedom offered by the shift parameter.

where $\boldsymbol{\sigma}^2$ denotes the vector of variances for the linear kernel, and $\boldsymbol{c}$ represents the shift parameter. Unlike the other primitive kernels, the linear kernel possesses independent variance terms for each input dimension.

The product between the linear kernel and the noise kernel generates a form of noise whose variance changes linearly with respect to the inputs. This presents an opportunity to model heteroscedastic noise, within the conventionally homoscedastic domain of GP regression models. When modelling real world tasks, this potentially offers a major advantage, since the noise variance often changes across the input space.

Note that if we simply set $c = 0$, as is often assumed when working with the linear kernel, then the linearly varying noise term is extremely limited. The noise variance could only ever increase as we move away from the origin. As with the cosine kernel, this reduces us to a small fraction $(2^{1-D})$ of the viable parameter space. In order to lift this restriction, we introduce a 'shift' vector in the linear kernel, such that the origin is free to move along each input dimension. This naturally leads us to ask what an appropriate prior for this shift vector would be. We consider it equally likely that the noise amplitude increases or decreases with $x$. To reflect this belief, and accounting for our normalised inputs, we seek a prior of the form $d\sigma^2/dx \sim \mathcal{N}(0, 0.1)$. For large displacements, we note that the shift parameter can be approximately expressed as the ratio of two normally distributed variables: the gradient of the noise and the noise amplitude at the origin. This observation suggests a suitable prior on $c$ is given by the Cauchy distribution. We adopt $c \sim \mathrm{Cauchy}(0, 5)$ throughout.

**Model Architecture.** The kernel's likelihood $p(\mathcal{D}|K)$ is invariant to permutations of the ordering of the datapoints, and to permutations of the ordering of the input dimensions. KITT's output of kernel recommendations should therefore exhibit these two important properties. Zaheer et al. [34] demonstrated that a function $f(X)$ which is invariant to permutations in the elements of $X$ can be expressed in the form $\rho\left(\sum_i \phi(x_i)\right)$, where $\rho$ and $\phi$ are differentiable functions. This was exploited by Lee et al. [18] in constructing the Set Transformer. The scenario we encounter is slightly more complex in that there is a two-tiered hierarchy of invariances. We seek a function over the training set $\mathcal{D}$, which can be expressed as a collection of high dimensional data vectors $D_i$: $f(\{\mathcal{D}\}) = \rho\left[\sum_i \phi(\mathcal{D}_i)\right]$. Here the function $\phi(\mathcal{D}_i)$ must be invariant over the permutations of the different input dimensions. This can therefore be decomposed in a similar manner, $\phi(D_i) = \rho'\left[\sum_j \phi'(D_{ij})\right]$. Combining these two equations leaves a final expression of the form

$$f(\{\mathcal{D}\}) = \rho\left(\sum_i \rho'\left(\sum_j \phi'(\mathcal{D}_{ij})\right)\right), \tag{4}$$

where $i$ and $j$ can be either dimensions or datapoints. This can be interpreted as: encode over $j$; pool over $j$; encode over $i$; pool over $i$; decode. This formalism sets the foundations for our choice of architecture, as shown in Figure 1. At a lower level, KITT is comprised of the following components whose acronyms we define here:

- rFF: Row-wise feed-forward layer with ReLU
- SAB: Set Attention Block [18]
- Multihead: Multi-Headed Attention Mechanism [29].
- MP: Mean pooling function
- LayerNorm: Layer Normalisation [1]

**Encoder:** The encoder has the architecture of a transformer with self attention blocks. Our goal is to encode datasets $\mathcal{D}_j = \{\boldsymbol{x}_i, y_i\}_{i=1}^N$ of shape $N \times (D + 1)$ where $\boldsymbol{x}_i \in \mathbb{R}^D$ and $y_i \in \mathbb{R}$; we need to incorporate both invariance to ordering of the data points (row-wise shuffle) and equivariance to a re-ordering of the dimensions (column-wise shuffle). In order to achieve this, our encoder has two sub-components responsible for encoding along the sequence and dimension axes respectively:

1. **SEQ_ENC:** $\mathbb{R}^{N \times (D+1)} \to \mathbb{R}^{D \times E}$. A sequence encoding component which acts on input datasets and outputs dimension-level representations $\mathrm{G} \equiv \{\boldsymbol{g}_d\}_{d=1}^D, \boldsymbol{g}_d \in \mathbb{R}^E$ where $E$ is the embedding dimension. The sequence encoder forward pass is formulated as:

$$\mathrm{SEQ\_ENC}(\mathcal{D}) = \mathrm{MP}(\mathrm{SAB}_{\times 6}(\mathrm{rFF}(\mathcal{D})))$$

    Where mean pooling is applied over the sequence. Our implementation of the SAB component [18] differs slightly from that of Lee et al. [18], $\mathrm{SAB}(Z) = \mathrm{LayerNorm}(C + Z)$, where we have defined $C = \mathrm{Dropout}(\mathrm{rFF}[\mathrm{Multihead}(Z, Z, Z)])$ [29].

2. **DIM_ENC:** $\mathbb{R}^{D \times E} \to \mathbb{R}^{D \times E}$ A dimension encoding component: DIM_ENC which acts on dimension level encodings to generate final representations $\{\boldsymbol{h}_d\}_{d=1}^D$ of dimension $E$. The dimension encoder forward pass is formulated as:

$$\mathrm{DIM\_ENC}(\mathrm{G}) = \mathrm{rFF}_{\times 2}(\mathrm{SAB}_{\times 6}(\mathrm{G}))$$

The encoder forward pass entails passing each input data set to the sequence encoding component followed by the dimension encoding component.

$$\mathrm{ENCODER}(\mathcal{D}) = \mathrm{DIM\_ENC}(\mathrm{SEQ\_ENC}(\mathcal{D}))$$

**Proposition 1:** *Let $\mathcal{S}_N$ denote the set of all permutations of the row-wise indices $\{1, 2, \ldots, N\}$ and $\mathcal{D}_\pi$ denote an input dataset with the ordering of indices given by $\pi \in \mathcal{S}_N$ The sequence encoding component SEQ_ENC is invariant to a permutation of the indices within a dataset $\mathcal{D}$.*

$$\mathrm{SEQ\_ENC}(\mathcal{D}) = \mathrm{SEQ\_ENC}(\mathcal{D}_\pi) \, \forall \, \pi \in \mathcal{S}_N$$

**Proposition 2:** *Let $\mathcal{Q}_D$ denote the set of all permutations of the column-wise indices (dimensions) $\{1, 2, \ldots, D\}$ and $\mathcal{D}_\nu$ denote an input dataset with the ordering of indices given by an arbitrary $\nu \in \mathcal{Q}_D$. The sequence encoder SEQ_ENC and dimension encoder DIM_ENC components are equivariant to a permutation of the dimensions within a dataset $\mathcal{D}$.*

$$\mathrm{ENCODER}(\mathcal{D}_\nu) = \nu(\mathrm{DIM\_ENC}(\mathrm{SEQ\_ENC}(\mathcal{D}))) \, \forall \, \nu \in \mathcal{Q}_D$$

We include proofs for propositions 1 and 2 in the supplementary.

**Decoder:** KITT's decoder iteratively builds a caption from the encoded dataset representations, generated by the encoder described above, and a prompt which consists of the kernel expression thus far (see Fig. 1). Our decoder closely resembles the one proposed in Vaswani et al. [29] except that we remove the positional encodings and adjust the number of layers. The decoder uses self-attention blocks to first attend to the prompt and then to attend to the representations from the encoder using the processed dataset representations as query values. These two applications of attention are alternated in several layers until a new component kernel is proposed from a distribution generated from the final representations. We note that this component kernel proposition is invariant to the ordering of the dataset representations and thus to a shuffling of the dimensions of the original dataset. Hence, the model is fully invariant. The end-to-end process is depicted in Figure 1 and a detailed schematic of the decoder is included in the supplementary material.

**Scalability.** Training of KITT is only performed once. Once training of the network is completed, all inference procedures require only a single forward pass through the network. As a result, instead of the $\mathcal{O}(N^3)$ cost commonly associated with explicit marginal likelihood evaluations, the cost of a forward pass through KITT is $\mathcal{O}(DN^2 + D^2)$. Due to effective parallelisation of the attention mechanism on GPUs, noted by Vaswani et al. [29], and the modest size of the KITT network, we experience near constant wall clock time in practice.

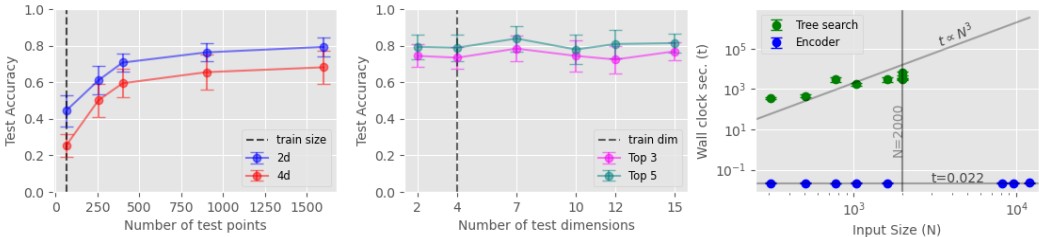

Figure 3: *Left and centre*: Classification performance for random samples drawn from primitive kernels across a range of test sizes and dimensionality. The vertical dashed lines denote the conditions under which the network was trained. *Right:* The time taken to predict a kernel for each of the UCI datasets. While KITT's overhead remains approximately constant, the tree search becomes impractical for larger inputs.

**Inference.** Given some new dataset $\mathcal{D}$, inference proceeds as below, closely mimicking the procedure used to generate a caption for images. Note that the best caption is not necessarily constructed by choosing the best kernel at each step of the decoder.

1. Pass the data $\mathcal{D}$ into the encoder; pass the resulting encodings and an (initially empty) kernel expression $\mathcal{E}$ into the decoder.

2. Retrieve the output probabilities, selecting the kernel $k$ with the highest probability and append the chosen kernel (or operator) to our full kernel expression $\mathcal{E}$.

3. Repeat steps 1 & 2 until either the $<\text{STOP}>$ token is selected or the maximum caption length is reached.

4. Repeat steps 1-3 several times but now select kernels stochastically, weighted by their probability, to construct a set of high-ranking kernel expressions.

5. For each of the top three candidate kernels expressions, as ranked by their total probability assigned by the network, we optimise the associated hyperparameters with BFGS [9].

6. Combine the posterior distribution of the forecasts based on either the overall probability assigned by the network, or another proxy for the model evidence such as the Bayesian Information Criterion.

In summary, we select a kernel based upon the output of the pretrained KITT network, before optimising the hyperparameters in a conventional manner.

## 4 Experimental Results

In this section we explore KITT's ability to predict kernels for synthetic data and standard regression benchmarks. We present four baselines to assess the performance of KITT on regression data. AHGP [19] is another deep network designed to assist GP inference, as outlined in 2. The neural kernel network [28], offers a differentiable form of kernel composition. We also include a greedy kernel search algorithm based upon the *Automatic Statistician* procedure outlined in Duvenaud et al. [7]. These three algorithms span a wide range of computational overheads, with AHGP being the fastest and the kernel search being the slowest. As a more familiar reference point, we also include the RBF-ARD baseline, which uses the same priors described previously.

**Ground Truth Recovery.** Identifying primitive structure is an important building block in being able produce sensible kernel recommendations for real-data in high dimensions. Capturing this structure is the aim of our encoder. In order to test the ability of the encoder to capture primitive structure, we form a classification transformer by taking the KITT encoder and appending a dense layer followed by a softmax activation. The resulting model is trained on datasets of fixed size and dimensionality to predict kernels for synthetic datasets drawn from GPs with known kernels. We demonstrate test performance in terms of accuracy for varying test input sizes and dimensions. We draw 300 random samples from a selection of primitive kernels, for varying combinations of test size and dimensionality. The results shown in Figure 3 demonstrate that the classifier is able to reliably

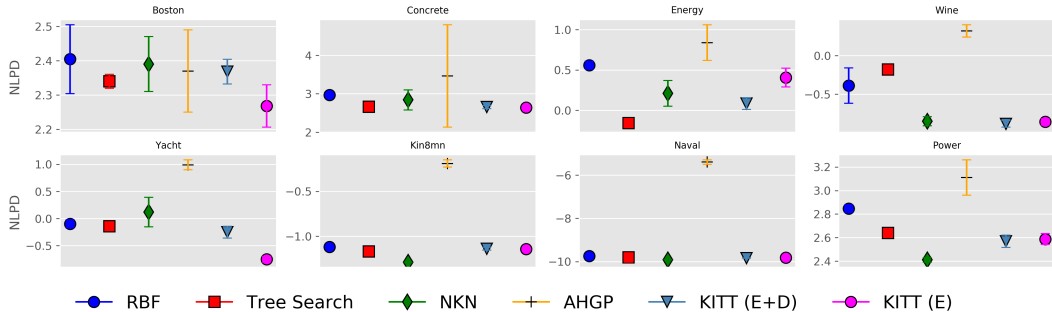

Figure 4: Negative predictive log likelihood values on UCI regression tasks for a variety of kernel selection methods. KITT remains competitive with the most computationally intensive approach, the tree search, while offering the advantage of being several orders of magnitude faster. The 'E' denotes sole use of the encoder followed by a classification layer to select kernels, while 'E+D' generates captions with the decoder.

generalise its structure detection capabilities to higher dimensional tasks, which were unseen during training of the network. As is expected, a moderately sized dataset of at least $\sim 200$ points is needed to achieve a reasonable level of prediction accuracy, and this continues to improve with increasingly large datasets.

**UCI Regression.** We evaluate KITT on eight real-world UCI regression tasks[2], spanning a range of input sizes and input dimensions (from 4 to 14). We adopt the same benchmarking methodology as Liu et al. [19], which includes a 90/10 train/test split, and subsampling 2,000 datapoints for those cases where the dataset exceeds this number. For each dataset, we predict a kernel caption with a maximum expression length of three terms. The caption is either constructed sequentially by the decoder, or in the case of the classifier, by summing the three highest scoring kernels.

The resulting NLPD values from KITT, and three other approaches to kernel design, are shown in Figure 4. Uncertainties are estimated from repeated experiments with ten different splits. KITT consistently outperforms the other transformer-based model, AHGP, and is competitive with the far slower tree search method (see Figure 3). We also perform an ablation study, details of which can be found in the supplementary material, demonstrating that KITT outperforms a random selection from its vocabulary.

We note that the AHGP performance is weaker than that of the RBF. This is significant because the RBF is a subset of the Spectral Mixture Product model, equivalent to a single component set to zero mean frequency. This suggests that the AHGP network perhaps focused on learning how to identify the lengthscale of the primary component of the multi-component Spectral Mixture Product.

For a deeper understanding of KITT's performance, Figure 5 compares the network's output against realised test performance on the Yacht dataset, across all 34 kernel classes. The three kernels KITT assigned high probability to, namely Linear $\times$ RBF, Linear $\times$ Matern32 and Linear $\times$ Matern52, correspond to the three strongest test performances.

**Computational overhead.** One of the most compelling features of KITT is the speed at which inference can be performed. Identifying a suitable kernel for a previously unseen dataset only entails a single forward pass through the encoder, and a small number through the decoder, each of which requires around two hundredths of a second. Furthermore, as illustrated in the right hand panel of Figure 3, the time-cost of prediction is robust to increasing data-set sizes and dimensionality.

The KITT network was trained on a Tesla V100 GPU for approximately eight hours, with Adam [16]. This procedure occurred only once, and does not need to be repeated when performing inference. To generate a kernel prediction requires a small fraction of a second, and is largely insensitive

---

[2]A recurring misconception in the literature is that the predictive errors on the Naval dataset are so small that they may be neglected, and these are sometimes listed as "0.00". We stress that they are small only because of the small variance of the raw data. This should have no bearing on its significance alongside the other datasets, and the RMSE should not be rounded down to zero.

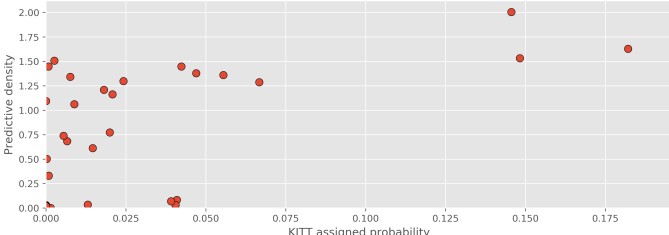

Figure 5: A comparison of KITT's kernel predictions against their test performance, on the Yacht dataset. Each dot represents one of the 34 kernels in KITT's vocabulary. KITT successfully identifies the three top performing kernels, and assigns low probability to the 31 alternative options.

to the size of the input data, as seen in the right hand panel of Figure 3. Once a kernel has been recommended, training typically required a further ten seconds. It is possible this step could also be greatly accelerated in future, if KITT were used in tandem with a hyperparameter optimisation network, similar to AHGP [19].

## 5 Related work

In this section we review approaches that either directly, through kernel construction, or indirectly target the issue of model selection in GPs.

**Amortised Hyperparmeter Learning (AHGP):** Liu et al. [19] also use self-attention based transformers, but with the goal of amortising hyperparameter learning. They train on input-output regression based datasets to estimate the final set of GP hyperparameters that would otherwise be learnt as a result of maximizing the marginal likelihood. In order to circumvent kerrnel selection they choose the flexible spectral mixture (SM) kernel with a fixed number of components per dimension, yielding a kernel with product structure over dimensions. The SM kernel arises from modelling the spectral density (Fourier dual) of a kernel function as a Gaussian mixture. Our work differs from AHGP as we focus on kernel design rather than optimising hyperparameter values.

**Kernel Engineering:** There are several examples in the GP literature of kernels being handcrafted to model one- or two-dimensional data [6, 7, 8, 24]. While this may be feasible in low-dimensional data with the aid of visual inspection, it is much less straightforward in a high-dimensional settings. Automated kernel engineering approaches search over a finite space of kernel structures which are progressively built by adding and multiplying a small number of base kernels. The focus is on devising an effective search algorithm over discrete structures where the end result is a composite kernel built from simpler known base kernels. This is largely the idea behind the *Automatic Statistician* project [15] where a greedy search procedure searches over all possible operators and sub-expressions to select the highest scoring combination. Our work similarly operates on a universe of compositional kernel structures but with a distinctly different model where we regress kernel labels on data sets with end-to-end gradient based training yielding a fast and scalable method.

**Deep Kernels:** There are other methods that bring to bear both the benefits of deep architecture and the analytical flexibility of kernel methods for the problem of representation learning [4, 13, 31]. The methods work by transforming the inputs to a GP with a neural network (NN) and jointly learning the parameters of the NN and the GP. The contention is that a simple base kernel (like a squared exponential (SE) kernel) works better when applied to the representations learnt by the NN than when applied to the raw input. These works try to side-step the problem of learning a sophisticated kernel apt for the data by focusing instead on learning a transformation of inputs. However, these methods can suffer from overfitting due to the joint training of millions of parameters of the NN in conjunction with the GP hyperparameters [20].

**Novel Kernels:** Other noted work includes the spectral mixture kernel which reparameterizes the kernel in terms of its spectral density (see *Bochner's Theorem* [3]) and derives closed form kernels which can be used as drop-in replacements for any stationary kernel function [26, 30].

# 6 Discussion

This work proposes a novel approach to addressing the kernel selection problem in GPs. By leveraging the potential for unlimited training data, we train a transformer-based model to identify the likelihood of a sample given a kernel class. Despite being trained solely on synthetic data, KITT is capable of selecting suitable kernels for previously unseen, real-world datasets. While we focus our efforts on the case of one-dimensional outputs, similar models could be developed for multi-output regression, classification, latent variable modelling and time-series prediction tasks. A major advantage of a pre-trained model for kernel structure detection is the speed of inference. By being able to recommend a kernel in a fraction of a second, KITT is dramatically faster than competing methods such as greedy search algorithms or differentiable kernel networks. Furthermore, it offers superior scalability. Empirically, we found that KITT is capable of pattern discovery across a broad range of input dimensions and dataset sizes. It was found to predict competitive kernels for high-dimensional real valued regression tasks. The ground truth experiments demonstrate its generalisation ability where it is able to identify structure in high-dimensional datasets. This work presents a powerful hybrid approach where kernel selection is informed by representation learning, by inferring a range of kernels compatible with the data. This achieves two aims of expressivity and ensemble uncertainty while spurring new possibilities for informed model selection in Gaussian processes. Given the high degree of complementarity with AHGP, which offers near-instantaneous optimisation of hyperparamters, there appears promising prospects for transformers to enhance the development, flexibility and scalability of Gaussian Process models.

## Acknowledgments and Disclosure of Funding

The authors would like to thank the anonymous reviewers for their helpful feedback. VL acknowledges funding from the Alan Turing Institute and Qualcomm Innovation Fellowship (Europe).

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
