# Supplementary Material for Kernel Identification Through Transformers

## A   Background: Self-Attention

Since the attention mechanism is rarely used within the GP literature, we provide a brief review of the topic in this section. Below we follow the description of attention as given by Vaswani et al. [8], including extensions to self-attention and multi-head self-attention.

The dot-product attention mechanism [8] takes as input a set of queries, keys and values. The queries and keys have dimension $D_z$ and the values have dimension $D_v$ which may differ from $D_z$. The operation of dot-product attention then generates weights from the queries and keys which are used to produce a linear mapping of the input values.

$$Attention(Q, K, V) = \text{softmax}\left(\frac{QK^\top}{\sqrt{D_z}}\right) V\,, \tag{5}$$

where the $Q$, $K$ and $V$ matrices denote the row-wise collection of queries, keys and values respectively. The softmax operation is applied row-wise with scaling to avoid the inputs exploding which hampers training. Intuitively, attention enables the input values to be processed to yield representations which account for the context of all other values in the sequence.

Dot-product self-attention acts on a single sequence of inputs, using it to generate queries, keys and values for the attention mechanism described above. Queries, Keys and Values are generated by right multiplication with learned weight matrices, $W_Q$, $W_K$ and $W_V$ respectively. Self-attention is therefore a mathematical operation which takes as input, a set of vectors of length $D$ collected row-wise in $Z$, $Z \equiv \{z_i\}_{i=1}^N, z_i \in \mathbb{R}^D$, and computes a weighted sum of vectors for each index $i$,

$$y_i = \sum_j w_{ij} z_j\,. \tag{6}$$

The weights, which collectively form a matrix $W$, are a function of the input vectors in $Z$,

$$W = \text{softmax}\left(\hat{W}_i/\sqrt{D}\right) \quad \text{where} \quad \hat{W} = ZW_Q W_K^\top Z^\top\,. \tag{7}$$

When this operation is repeated $H$ times with individual sets of matrices $\{W_Q^h, W_K^h, W_V^h\}_{h=1}^H$ the resulting operation is called *Multi-Head Self Attention* (MHSA). Let $a_h$ denote the attention encoded output of a single head, the output of multi-head attention is then computed as,

$$\text{MHSA}(Z) \equiv \text{Multihead}(Z, Z, Z) = \underbrace{\text{concat}(a_1, ...a_H)}_{N \times HD} W_0 \in \mathbb{R}^D\,, \tag{8}$$

$$\text{where } a_h = Attention_h(ZW_Q^h, ZW_K^h, ZW_V^h)\,. \tag{9}$$

Here $W_0 \in \mathbb{R}^{HD \times D}$ is the final trainable projection matrix which brings the outputs of the individual heads back to the original dimension $D$.

35th Conference on Neural Information Processing Systems (NeurIPS 2021).

## B Proofs

### B.1 On encoding sets of datasets

Our transformer architecture is trained on input-output datasets where the output is drawn from GP priors with a known kernel. Each labeled example is thus a self-contained dataset. This introduces some challenges in ensuring that each dataset's encoding is not sensitive to the ordering of the data points within each dataset . A training instance $\mathcal{D} = \{\{\boldsymbol{x}_{i,j}\}_{j=1}^{D}, y_i\}_{i=1}^{N} \in \mathbb{R}^{N \times D+1}$ denotes a rank-2 input tensor which carries the interpretation of a **set** of $N$ vectors in $\mathbb{R}^{D+1}$. For instance, $[x_{i,1}, \ldots, x_{i,D}, y_i]$ is the $i^{th}$ vector. Hence, we wish to preserve invariance of the encoded representation with respect to a permutation of the rows within each dataset.

$$\text{ENCODER}(\mathcal{D}_{\pi}) = \text{ENCODER}(\mathcal{D}) \tag{10}$$

where $\mathcal{D}_{\pi} = \{\{\boldsymbol{x}_{\pi(i),j}\}_{j=1}^{D}, y_i\}_{i=1}^{N}$, and $\pi : \{1, \ldots, N\} \to \{1, \ldots, N\}$ is a bijective permutation function on the row indices. This is addressed by Proposition 1.

Further, we want to ensure that the encodings of each dataset are equivariant to the ordering of the input dimensions (columns) in each dataset. If $\nu : \{1, \ldots, D\} \to \{1, \ldots, D\}$ is a bijective permutation function on the column indices then,

$$\text{ENCODER}(\mathcal{D}_{\nu}) = \nu(\text{ENCODER}(\mathcal{D})) \tag{11}$$

where $\mathcal{D}_{\nu} = \{\{\boldsymbol{x}_{i,\nu(j)}\}_{j=1}^{D}, y_i\}_{i=1}^{N}$. Note that the position of dimension $(D+1)$ denoting the output column in each training instance is always preserved. We address this permutation equivariance in Proposition 2.

**Lemma 1:** The self-attention mechanism (for each head) defined in A is permutation equivariant.

$$\text{softmax}\left(\frac{Z_{\pi}W_Q W_K^T Z_{\pi}^T}{\sqrt{D}}\right) Z_{\pi}W_V = \pi\left(\text{softmax}\left(\frac{ZW_Q W_K^T Z^T}{\sqrt{D}}\right) ZW_V\right) \forall \pi \in S_N$$

where $Z_{\pi} = \pi(Z)$ denotes a permutation of the rows in $Z, \pi \in S_N$ where $S_N$ denotes the set of all permutations of the row indices $\{1, \ldots, N\}$.

*Proof.* First, we note that the softmax and scaling are element-wise operations and don't interfere with the ordering of rows; in order to prove the permutation equivariance we just need to focus on the matrix multiplication operations.

To prove:

$$Z_{\pi}J Z_{\pi}^T Z_{\pi}W_V = \pi(ZJZ^T ZW_V) \tag{12}$$

where we have pre-multiplied $W_Q W_K^T = J$ (by the associativity of matrix multiplication).

Without loss of generality assume,

$$Z = \begin{bmatrix} a & b \\ c & d \end{bmatrix} \quad Z_{\pi} = \begin{bmatrix} c & d \\ a & b \end{bmatrix} \quad Z_{\pi}^T = \begin{bmatrix} c & a \\ d & b \end{bmatrix} \quad J = \begin{bmatrix} j_1 & j_2 \\ j_3 & j_4 \end{bmatrix} \quad W_V = \begin{bmatrix} v_1 & v_2 \\ v_3 & v_4 \end{bmatrix} \tag{13}$$

First, we note that the operation $Z^T Z$ is permutation invariant, $Z_{\pi}^T Z_{\pi} = Z^T Z$

$$Z_{\pi}^T Z_{\pi} = \begin{bmatrix} c & a \\ d & b \end{bmatrix}\begin{bmatrix} c & d \\ a & b \end{bmatrix} = \begin{bmatrix} c^2 + a^2 & cd + ab \\ cd + ab & b^2 + d^2 \end{bmatrix} = \begin{bmatrix} a & c \\ b & d \end{bmatrix}\begin{bmatrix} a & b \\ c & d \end{bmatrix} = Z^T Z$$

Hence, the LHS term in (12) becomes,

$$Z_{\pi}J Z_{\pi}^T Z_{\pi}W_V = Z_{\pi}\underbrace{JZ^T ZW_V}_{H} = Z_{\pi}H = \pi(ZH) \tag{14}$$

where the final equality can be shown to be true by assuming without loss of generality,

$$H = \begin{bmatrix} h_1 & h_2 \\ h_3 & h_4 \end{bmatrix},$$

Hence,

$$Z_\pi H = \begin{bmatrix} c & d \\ a & b \end{bmatrix} \begin{bmatrix} h_1 & h_2 \\ h_3 & h_4 \end{bmatrix} = \begin{bmatrix} ch_1 + dh_3 & ch_2 + dh_4 \\ ah_1 + bh_3 & ah_2 + bh_4 \end{bmatrix}$$
$$= \pi \left( \begin{bmatrix} ah_1 + bh_3 & ah_2 + bh_4 \\ ch_1 + dh_3 & ch_2 + dh_4 \end{bmatrix} \right)$$
$$= \pi \left( \begin{bmatrix} a & b \\ c & d \end{bmatrix} \begin{bmatrix} h_1 & h_2 \\ h_3 & h_4 \end{bmatrix} \right)$$
$$= \pi(ZH)$$

$\square$

Note that this proof shows equivariance of self-attention in a rank 2 input tensor case ($Z \in \mathbb{R}^{D \times D}$) however the proof can be generalised to higher rank inputs. For example, if $Z \in \mathbb{R}^{D \times E \times J}$ is a rank 3 input tensor and we permute the indices on the $E$ dimension $Z_{D \times E \times J} \to Z_{D \times \pi(E) \times J} = Z_{\pi^E}$, then, the output of self-attention is permuted on the same dimension.

$$\text{MHSA}(Z_{\pi^E}) = \pi^E(\text{MHSA}(Z)) \tag{15}$$

In the full encoder architecture, the attention mechanism is applied internally as part of a sequence of blocks called set attention blocks SAB[5][1]. The block operation leaves the architecture permutation equivariant.

**Lemma 1.1:** *The set attention block* $\text{SAB}(Z)$ *is permutation equivariant.*

$$\text{SAB}(Z) := \text{LayerNorm}(C + Z)$$

where C denotes a context vector computed with the Attention Mechanism A.

$$C = \text{rFF}(\text{MHSA}(Z))$$

*Proof.* We know that the MHSA($\cdot$) operation is permutation equivariant and LayerNorm is an independent element-wise operation with no parameters. It remains to verify that the feed-forward operation leaves the outputs permutation equivariant.

Without loss of generality assume that the outputs of MHSA($Z$) are given by an $N$-$D$ tensor $A \in \mathbb{R}^{D \times D \ldots \times D}$. The feed-forward layer with $k$ hidden units applies a matrix $W_k$ of dimension $D \times k$ along the last axis of the inputs yielding a tensor output rFF($A$) $= AW_k$ of shape $D \times \ldots D \times k$. Essentially, each sub-tensor of shape $(1 \times \ldots \times D)$ (row of size $D$) is multiplied by the weight matrix independently and identically to yield the output sub-tensors of shape $(1 \times \ldots 1 \times k)$ (row of size $k$). Since this operation is applied row-wise it is permutation equivariant to the order of the order of the sub-tensors in $A \Rightarrow \text{rFF}(A_\pi) = \pi(\text{rFF}(A))$. $\square$

**Proposition 1:** *Let* $\mathcal{S}_N$ *denote the set of all permutations of the row-wise indices* $\{1, 2, \ldots, N\}$ *and* $\mathcal{D}_\pi$ *denote an input dataset with the ordering of indices given by* $\pi \in \mathcal{S}_N$ *The sequence encoding component* SEQ_ENC *is invariant to a permutation of the indices within a dataset* $\mathcal{D}$.

$$\text{SEQ\_ENC}(\mathcal{D}) = \text{SEQ\_ENC}(\mathcal{D}_\pi) \ \forall \ \pi \text{ in } \mathcal{S}_N$$

*Proof.* The sequence encoder forward pass is formulated as:

$$\text{SEQ\_ENC}(\mathcal{D}) = \text{MP}(\text{SAB}_{\times 6}(\text{rFF}(\text{R}(\mathcal{D}))))$$

where $R$ is a reshape operation which takes a rank-2 tensor input of size $(N \times D + 1)$ and outputs a rank-3 tensor of shape $(D \times N \times 2)$. This reshaped tensor is formed by stacking row-wise of the $N \times 2$ sub-tensors corresponding to each dimension yielding $\{\{(x_{i,j}, y_i)\}_{i=1}^N\}_{j=1}^D$. Let $\mathcal{D}_\pi$ denote

---

[1] We note that our implementation of the set attention block differs slightly in its use of dropout and residual connections from that of Lee et al. [5].

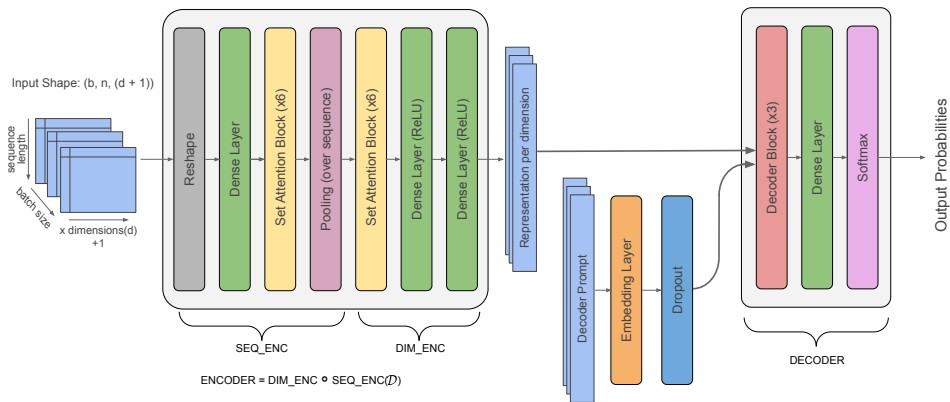

Figure 6: KITT architecture adapted from image captioning network Xu et al. [9].

a training instance (a dataset) of shape $(N \times D + 1)$ where $\pi$ denotes a permutation of the rows. The output of the reshape operation $R$ is a rank-3 input tensor of shape $D \times \pi(N) \times 2$ where the order of data points has been permuted for each dimension. rFF is permutation equivariant as it acts on rows of the input dataset, hence $\text{rFF}(\text{R}(\mathcal{D}_\pi)) = \pi(\text{rFF}(\text{R}(\mathcal{D})))$. The SAB block [5] is permutation equivariant from Lemma 1.1. We know that permutation equivariant layers stacked together are permutation equivariant [10]. Hence, a composition of SAB layers ($\text{SAB}_{\times 6}$) with rFF is permutation equivariant.

Further, the mean-pooling operation MP applied across the sequence $(N)$ in each dimension is permutation invariant by definition, hence,

$$\text{SEQ\_ENC}(\mathcal{D}_\pi) = \text{MP}(\text{SAB}_{\times 6}(\text{rFF}(\text{R}(\mathcal{D}_\pi)))) \tag{16}$$

$$= \text{MP}(\pi(\text{SAB}_{\times 6}(\text{rFF}(\text{R}(\mathcal{D}))))) \tag{17}$$

$$= \text{MP}(\text{SAB}_{\times 6}(\text{rFF}(\text{R}(\mathcal{D})))) \tag{18}$$

$$= \text{SEQ\_ENC}(\mathcal{D}) \tag{19}$$

$$\square$$

**Proposition 2:** *Let $\mathcal{Q}_D$ denote the set of all permutations of the column-wise indices (dimensions) $\{1, 2, \ldots, D\}$ and $\mathcal{D}_\nu$ denote an input dataset with the ordering of indices given by $\nu \in \mathcal{Q}_D$. The sequence encoder* SEQ_ENC *and dimension encoder* DIM_ENC *components are equivariant to a permutation of the dimensions within a dataset $\mathcal{D}$.*

$$\text{ENCODER}(\mathcal{D}_\nu) = \nu(\text{DIM\_ENC}(\text{SEQ\_ENC}(\mathcal{D}))) \; \forall \; \nu \in \mathcal{Q}_D$$

*Proof.* First, we tackle the sequence encoder SEQ_ENC. Let $\mathcal{D}_\nu$ denote a training instance of shape $(N \times D + 1)$ where $\nu$ denotes a permutation of the input columns[2]. The output of the reshape operation is a 3d input tensor of shape $(D \times N \times 2)$. Since the feed-forward layer rFF applies to each $N \times 2$ sub-tensor independently and identically it renders the output permutation equivariant, $\text{rFF}(\text{R}(\mathcal{D}_\nu)) = \nu(\text{rFF}(\text{R}(\mathcal{D})))$.

The $\text{SAB}_{\times 6}$ block is permutation equivariant from Lemma 1.1. The mean-pooling operation collapses the dimension of size $N$ (i.e. it is applied across the sequence of data points in each dimension) yielding an output which is permutation equivariant to the order of dimensions.

Hence, $\text{SEQ\_ENC}(\mathcal{D}_\nu) = \nu(\text{SEQ\_ENC}(\mathcal{D}))$.

The dimension encoder DIM_ENC is implemented as a stack of multi-head attention blocks $\text{SAB}_{\times 6}$ which are permutation equivariant (as shown), hence the full encoder is a composition of permutation equivariant transformations (w.r.t. dimensions),

---

[2]The output column is always on the last axis

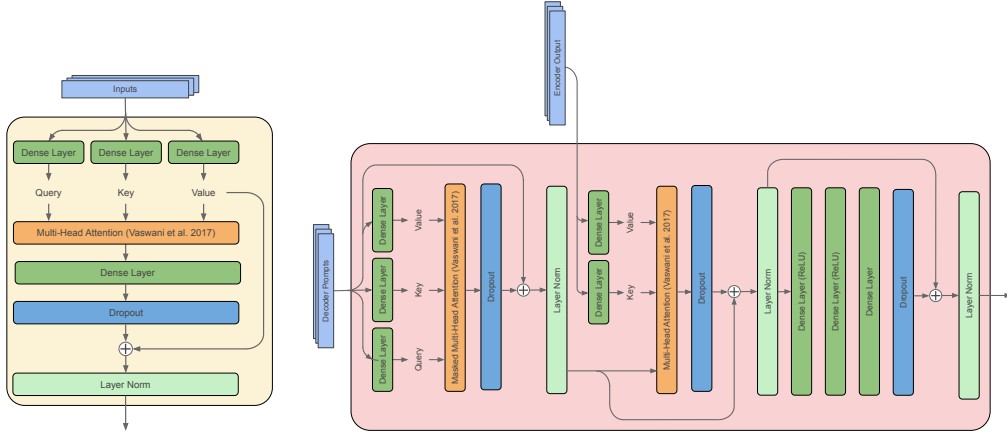

Figure 7: *Left:* Set Attention Block. *Right:* Decoder Block

$$\mathrm{DIM\_ENC}(\mathrm{SEQ\_ENC}(\mathcal{D}_\nu)) = \nu(\mathrm{DIM\_ENC}(\mathrm{SEQ\_ENC}(\mathcal{D})))$$

$\square$

The encodings produced by our encoder are therefore invariant to the ordering of data points and equivariant with respect to permutations of dimension. When coupled with the fact that we do not add any location encoding before passing the encodings to our decoder, this renders the kernels proposed by the full KITT model fully invariant to permutations of both the dimensions and indices in the input dataset. This follows from the decoder treating all of the provided encodings equally, attending to them according to the values of the queries generated by the prompt provided and the keys calculated from the encodings themselves. More information on the architecture of the decoder is provided in the next section.

## C   KITT Architecture

In this section, we expand on the discussion of KITT's architecture in the main body of the paper with additional detail and diagrams.

Figure 6 shows the full end-to-end KITT architecture which maps from a dataset to a distribution over kernels. Both the encoder and the decoder utilise sets of 6 attention blocks in conjunction with typical densely connected layers. Where no activation function is specified the dense layer simply represents multiplication by a matrix of learned weights. These dense layers operate only on the final dimension of the input tensors and are therefore referred to as row-wise feed forward layers (rFF). As shown in Figure 1, the operation of the decoder is recurrent in so far as it is run several times with the same encodings, updating the prompt with the proposed kernel from the last step at each iteration. We only run the decoder multiple times, as it is only the prompt that changes each time; once calculated the encoded dataset representations are cached and reused in each pass of the decoder. Note that during training there is no need for repeated forward passes through the decoder as the full caption is known. We are therefore able to train the decoder by passing copies of the caption with differing masking through the decoder to effectively parallelise decoder prediction at each point of caption generation. The mask acts to prevent looking ahead to what is to be predicted.

The internal operations of the Set Attention Blocks and the Decoder Block which form the core of the encoder and decoder respectively are depicted in Figure 7.

Our Set Attention Blocks differ slightly from those introduced by Lee et al. [5]. Following Liu et al. [6], we utilise dropout [4] with a rate of $0.1$, and favour a single residual connection and layernorm [2] operation as compared to the two instances of layer norm with residual connection used by Lee et al. [5].

A key difference between our encoder and the original transformer [8] is the absence of an explicit positional encoding added to the inputs. In our setting of processing full datasets, this is not required

| Approach | Kernel | Hyperparameters | Reference |
|---|---|---|---|
| Tree Search (TS) | Flexible | ML-II | Duvenaud [3] |
| Neural Kernel Network (NKN) | Flexible | ML-II | Sun et al. [7] |
| Amortised Hyper. Inference (AHGP) | Fixed (SM) | Amortised | Liu et al. [6] |
| KITT (This work) | Flexible | ML-II | – |

Table 1: A summary of the kernel selection and optimisation methods under consideration.

| Dataset | RBF | TS | NKN | AHGP | KITT (E + D) | KITT (E) |
|---|---|---|---|---|---|---|
| Boston | 2.40 (0.05) | 2.08 (0.02) | 2.39 (0.08) | 2.37 (0.12) | 2.37(0.04) | 2.27 (0.06) |
| Concrete | 2.96 (0.03) | 2.46 (0.03) | 2.84 (0.26) | 3.46 (1.33) | 2.65 (0.04) | 2.64 (0.04) |
| Energy | 0.56 (0.04) | -0.14 (0.05) | 0.21 (0.16) | 0.84 (0.22) | 0.08 (0.07) | 0.41 (0.12) |
| Wine | -3.94 (0.52) | -3.94 (0.05) | -0.85 (0.06) | 0.32 (0.08) | -0.88 (0.04) | -0.86 (0.04) |
| Yacht | -0.34 (0.02) | -0.40 (0.03) | 0.12 (0.27) | 0.99 (0.09) | -0.25 (0.11) | -0.75 (0.06) |
| Kin8mn | -1.12 (0.01) | -1.29 (0.02) | -1.29 (0.00) | -0.19 (0.04) | -1.14 (0.02) | -1.14 (0.02) |
| Naval | -9.74 (0.02) | -8.87 (0.22) | -9.92 (0.00) | -5.40 (0.10) | -9.84 (0.01) | -9.82 (0.01) |
| Power | 2.84 (0.03) | 2.33 (0.03) | 2.41 (0.02) | 3.11 (0.15) | 2.57 (0.05) | 2.59 (0.05) |

Table 2: A comparison of kernel learning approaches for UCI benchmarks. NLPD ($\pm$ standard error of mean) evaluated on average of 10 splits with 90% of the data used for training.

as the relevant positional information is naturally contained within the inputs. The position of a datapoint is defined by its $x$ values which form an integral part of the input and are therefore processed directly by the transformer without any need for external processing. This denotes an important generalisation of the Transformer beyond it's original application in natural language processing to our setting of detecting patterns in any numerical dataset.

Our decoder blocks are implemented as originally proposed by Vaswani et al. [8] when introducing the Transformer architecture. However, unlike Vaswani et al., we omit positional encodings from the input prompt. Positional encodings are not required when processing a kernel caption due to the transitivity of the addition of kernels ($k_a + k_b = k_b + k_a$). The order of previously predicted kernels does therefore not influence the prediction of the next kernel. It is only the set of previous predictions which is important. In order to enable the ordering of kernels within a sum to convey some degree of information, we adopt a formalism where kernels are stated in decreasing order of their variance.

## D    Further Experimental Results

Here we present further details of our experimental results. The NLPD values illustrated in Figure 4 in the main text, corresponding to the UCI regression tasks, are shown in Table 2. The RMSE values for the same experiments are given in Table 3. Finally, Table 4 shows that performing model averaging across the top three kernels offers some advantage over simply selecting the top one, while both approaches comfortably outperform a random kernel selection.

In all cases, quoted uncertainties are estimated from repeating ten different splits.

## E    Priors

An important step in the construction of the training data is defining a suitable set of priors for the hyperparameters for each kernel in the vocabulary. We select priors for each individual hyperparameter of the component kernels ensuring a wide support as discussed in the main text. For product kernels we learn a single variance hyperparameter, for instance, $\sigma_f^2(k_1 k_2 k_3)$ but each additive term has its own variance hyperparameter. Since the priors are assigned on individual lengthscales in each component kernel, the prior on the implied lengthscale ends up having compressed support over a narrower range of shorter lengthscales (see figure 8). Hence, the priors for the component lengthscale hyperparameters in product kernels are scaled to ensure a target prior over the implicit lengthscale. This is an important correction to make, because otherwise the network will learn to identify product kernels based upon their characteristically shorter lengthscales.

| Dataset | RBF | TS | NKN | AHGP | KITT (E + D) | KITT (E) |
|---|---|---|---|---|---|---|
| Boston | 3.06 (0.21) | 3.12 (0.29) | 2.51 (0.15) | 2.73 (0.38) | 3.14 (0.27) | 2.56 (0.13) |
| Concrete | 4.89 (0.17) | 3.83 (0.18) | 3.69 (0.24) | 3.45 (0.45) | 3.75 (0.18) | 3.79 (0.2) |
| Energy | 0.43 (0.02) | 0.28 (0.01) | 0.25 (0.02) | 0.51 (0.07) | 0.26 (0.012) | 0.28 (0.012) |
| Wine | 0.65 (0.01) | 0.55 (0.01) | 0.52 (0.01) | 0.58 (0.04) | 0.546 (0.008) | 0.545 (0.0074) |
| Yacht | 0.22 (0.02) | 0.22 (0.01) | 0.31 (0.06) | 0.46 (0.27) | 0.227 (0.024) | 0.187 (0.014) |
| Kin8mn | 0.08 (9e-04) | 0.08 (0.00) | 0.07 (0.00) | 0.19 (0.01) | 0.078 (0.001) | 0.078 (0.0011) |
| Naval | 1e-05 (5e-07) | 0.00 (0.00) | 0.00 (0.00) | 0.00 (0.00) | 1.58e-5 (5e-7) | 1.6e-5 (3.5e-7) |
| Power | 4.13 (0.12) | 3.33 (0.16) | 2.68 (0.07) | 4.23 (0.24) | 3.33 (0.17) | 3.36 (0.16) |

Table 3: A comparison of kernel learning approaches for UCI benchmarks. RMSE ($\pm$ standard error of mean) evaluated on average of 10 splits with 90% of the data used for training. The acronyms used here are defined in Table 1.

| Dataset | Random | KITT Top 1 | KITT Top 3 |
|---|---|---|---|
| Boston | 3.48 (0.56) | 3.24 (0.18) | 2.56 (0.13) |
| Concrete | 5.75 (0.91) | 4.03 (0.21) | 3.79 (0.2) |
| Energy | 1.43 (0.93) | 0.386 (0.012) | 0.28 (0.012) |
| Wine | 0.66 (0.02) | 0.55 (0.0077) | 0.545 (0.0074) |
| Yacht | 2.15 (0.95) | 0.209 (0.016) | 0.187 (0.014) |
| Kin8mn | 0.14 (0.02) | 0.08 (9e-04) | 0.078 (0.0011) |
| Naval | 0.01 (0.01) | 1.68e-05 (3.7e-07) | 1.6e-05 (3.5e-07) |
| Power | 5.43 (1.26) | 3.44 (0.14) | 3.36 (0.16) |

Table 4: Predictive performance (RMSE) for three different model averaging strategies.

**Remark 1:** The product of two RBF kernels with identical lengthscales, $l_1 = l_2$, yields another RBF kernel with lengthscale $l_{prod} = \dfrac{l_1}{\sqrt{2}}$. The product of three RBF kernels with identical lengthscales, $l_1 = l_2 = l_3$, yields another RBF kernel with lengthscale $l_{triple} = \dfrac{l_1 l_2 l_3}{\sqrt{l_1^2 + l_2^2 + l_3^2}}$.

**Remark 2:** If lengthscales $l_1, l_2, l_3 \sim \mathcal{LN}(\mu, \sigma^2)$ are independent log-normal random variables, then their product $l_1 l_2 l_3 \sim \mathcal{LN}(3\mu, 3\sigma^2)$ and $\sqrt{(l_2 l_3)^2 + (l_1 l_3)^2 + (l_1 l_2)^2} \dot{\sim} \mathcal{LN}(0.5\mu_z, 0.25\sigma_z^2)$ where,

$$\sigma_z^2 = \ln\left[\frac{\sum e^{2\mu+\sigma^2}(e^{\sigma^2}-1)}{(\sum e^{\mu+\sigma^2/2})^2} + 1\right], \tag{20}$$

$$\mu_z = \ln\left[\sum e^{\mu+\sigma^2/2}\right] - \frac{\sigma_z^2}{2}, \tag{21}$$

is an approximation to the sum of log-normally distributed random variables [1].

Hence, $l_{triple} \dot{\sim} \mathcal{LN}(3\mu - 0.5\mu_z, 3\sigma^2 + 0.25\sigma_z^2 - 2\rho(\sqrt{3}\sigma)(\sqrt{0.25}\sigma_z))$ where $\rho$ is the correlation coefficient. Since we can deduce how the implied lengthscales in product kernels are approximately distributed, we can compute an approximate scaling factor for the priors in each of the component kernels such that the overall shrinkage is compensated for. This corection is shown in Figure 9. The propagation of lengthscales will differ slightly for kernels whose spectral densities are non-Gaussian, however this is a relatively small impact.

# F   Experimental Details

## F.1   Ground-Truth Experiments

For each combination of test-size and dimensionality, we draw 300 GP samples from known primitive kernels and report test time classification accuracy. We report accuracy in terms of the fraction (%) of samples classified to the ground truth primitive kernel (i.e. correct classifications). We report

| Dataset | $N$ | $d$ | TS | KITT (E) |
|---------|-----|-----|-----|----------|
| Boston | 506 | 13 | 450.3 (32.6) | 0.023 (0.001) |
| Concrete | 1030 | 8 | 1701.2 (121.4) | 0.023 (0.001) |
| Energy | 768 | 8 | 2936.8 (364.1) | 0.023 (0.001) |
| Wine | 1599 | 11 | 2988.0 (409.5) | 0.022 (0.001) |
| Yacht | 308 | 6 | 352.9 (20.3) | 0.023 (0.002) |
| Kin8mn | 8192 | 8 | 3392.9 (104.7) | 0.022 (4e-04) |
| Naval | 11934 | 14 | 6194.2 (677.8) | 0.022 (4e-04) |
| Power | 9568 | 4 | 3174.1 (166.8) | 0.022 (4e-04) |

Table 5: Time taken (in seconds) to select a kernel, on an Intel i7-8700 12 CPU cores (3.20GHz, 32GB RAM) with one GPU (GTX 1070, 8GB RAM). We report means with 1 std. deviation estimated from 10 repetitions. For the Tree Search (TS) times, only a maximum of $N = 2,000$ datapoints are used.

means and standard errors over three runs per combination. Further, we report top-3 accuracy for the test size experiment and top-3/top-5 accuracy for the test dimensions experiment where we fixed the size of test inputs to $N = 1600$ across dimensions.

### F.2 ML-II Training

For all experiments, the initial set of hyperparameter values was determined by selecting the highest marginal likelihood from $1,000$ random draws from the priors.

## G Broader Impact

Selecting a kernel for a dataset to be modelled with a GP is a long-standing challenge and research goal for GP researchers. Techniques to learn the functional form of a kernel are usually extremely computationally intensive and can deter GP practitioners from using them. The common approach is to use a reasonable default and focus ones efforts on scalable training once a kernel has been fixed. We believe that a pre-trained solution such as the one we propose in our paper which removes the burden of expensive kernel selection at experiment time may enjoy wide-spread adoption and popularity. A sufficiently large pre-trained network available for direct download and inference enables users to cross-check and compare their hand-crafted kernels with automated proposals.

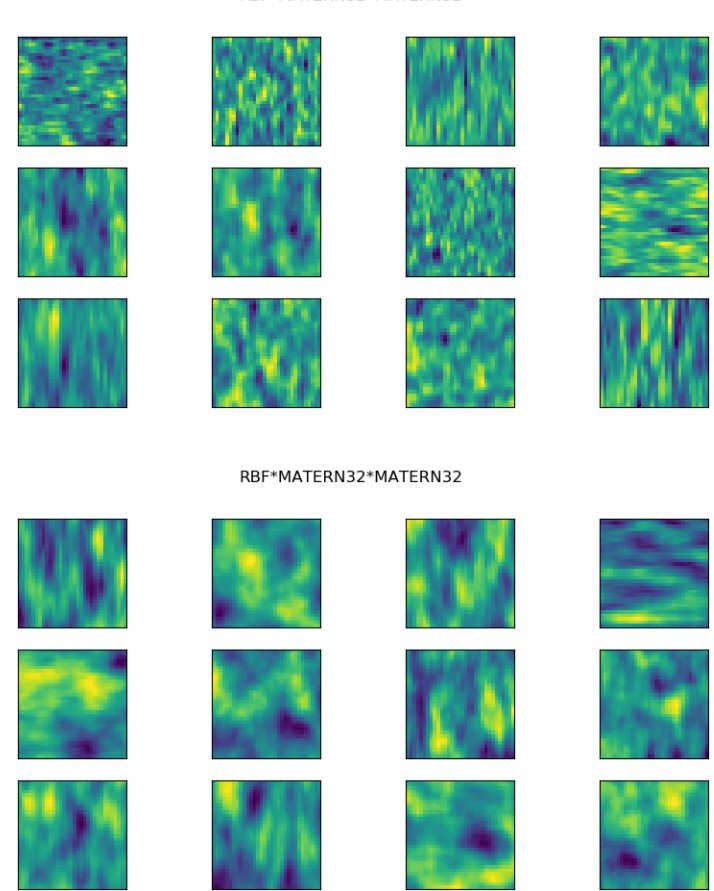

Figure 8: Random samples drawn from a triple product kernel across two input dimensions - before (top) and after (bottom) accounting for the induced shrinkage in lengthscale.

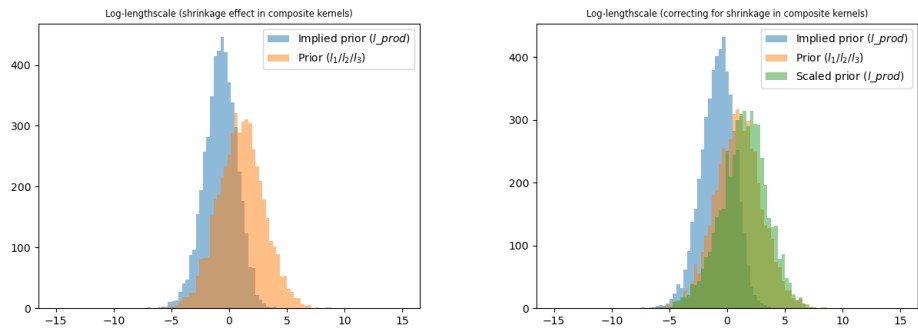

Figure 9: *Left:* Histogram of samples from the lengthscale prior distribution and the implied prior distribution for a product of three RBF kernels. *Right:* Additionally, samples (green) from the scaled prior ensuring that the range of lengthscales in the final product kernel has the same width as the component kernels.