# OpenReview forum: "Kernel Identification Through Transformers"
_NeurIPS.cc/2021/Conference — NeurIPS 2021 Poster_

### Official Review · Reviewer_VZVF · 2021-07-14

**Rating:** 6
**Confidence:** 5

**Summary:**

The paper presents a new approach to select kernel functions via transformers. The proposed approach mimics image captioning tasks where the model takes the pair of data covariates and targets as image inputs,and then produces kernel grammars as captions. The paper designs a transformer architecture in the way that it is invariant to permutations across input indices and input dimensions.

**Ethics Review Area:**

["I don’t know"]

**Limitations And Societal Impact:**

While experiments show promising results in medium-dimensional datasets, the paper may need to verify that the method should work for 1D datasets. Because it is easier to justify the quality of found kernels from plotting and visualization. A plot of kernel structure found by KITT for Mauna or airline passengers might be sufficient. One suggestion here is that the paper may follow the setup of the signal-to-noise experiment in [Duvenaud et al. 2013] to check if the method is sensitive to noise.

Minors:
- The explanation of why AHGP performs worse than RBF is not clear. If RBF is the subset of Spectral Mixture (SM), SM is at least as good as RBF. And if AHGP really learned the lengthscale of the primary component, the paper should provide a way to demonstrate it.

- Missing related work on stochastic grammar using probabilistic programming. For example, https://arxiv.org/pdf/1611.07051.pdf

Questions:

- The output of KITT is a sum of product kernels in which components are also commutative and order does not matter. Transformers for natural language generates sentences where the order or words within a sentence matters. Does it cause any difficulty to identify the kernel in KITT?

- What potential challenges may arise if data and kernel grammar includes change-point and change-window?

- In Figure 3 (first), why does the increase in the number of test points result in better classification accuracy although the model trained on 64 points?

- Is automatic relevant determination (ARD) considered during sampling from priors? Is there a way to extend this work where kernel structures vary across input dimensions?

---- After rebuttal ----
Overall, the proposed approach for GP kernel selection is interesting. So I raised the score to 6. However, the paper representation can be improved. I hope the authors could add the discussion of 1-dimensional case to the paper as in the rebuttal.


**Main Review:**

Kernel selection is one of the important problems in learning Gaussian processes. And using the representation power of transformers is a very interesting idea. The method allows to train a transformer once and reuse many times to predict kernel structures  and therefore avoids exhausted search procedures like in the automatic statistician.

However, the technical contribution of this paper is limited. The double permutation invariance is obtained via a function composition of deep sets. Although experiments show promising results, the paper may need more ablation study to demonstrate the efficacy of the proposed method (please see below).


**Time Spent Reviewing:**

12

---

> ### Author Response · Authors · 2021-08-10
> **Reply to Reviewer VZVF**
>
> Thank you for your interesting suggestion of exploring time series data. Unfortunately the preferred kernels in this regime tend to involve nested sums within products (eg the suggested Airline data as modelled by Duvenaud et al involves a kernel of the form SE × ( Lin + Lin × ( Per + RQ ) ) ). These expressions are inaccessible to the current kernel grammar, so we would not expect to find particularly good performance on one-dimensional datasets with the current setup. We certainly think that a KITT-like network could be made for identifying kernels for one-dimensional data, albeit with a different choice of architecture, since there is no need for dimensional invariance.
>
> While it is true that additive kernels are not ordered like words in a sentence, to facilitate training our caption are defined with a meaningful order - they are given in decreasing order of the variance of its component kernels.
>
> The suggestion of including the change-point and change-window operators is an intriguing one, and we believe it represents a promising avenue for future exploration. Their drawback is they can prove very difficult to optimise, since they tend to be associated with highly multi-modal likelihoods. We believe these operators could therefore be well suited for a future network that combines KITT and AHGP-like abilities, as this would bypass the need for gradient descent.
>
> We also agree that it would be interesting to explicitly demonstrate that the weights of AHGP's outputs are dominated by a single component, to confirm the hypothesis that it's essentially learning an RBF. However we feel this exploration lies outside the scope of the current work. We believe AHGP represents an exciting first step in radically improving the speed of hyperparameter optimisation, just as KITT represents an exciting first step in drastically accelerating the process of kernel selection.
>
> Increasing in the number of test points (here we refer to test points from the perspective of kernel selection) provides the network with more information, so it becomes easier to distinguish different kernel structure in a given sample. This holds true despite KITT being trained on samples of 64 points thanks to the ability of transformers to generalise to longer sequences than those trained upon - this is an important aspect of the architecture that we perhaps ought to have highlighted more clearly.
>
> We can confirm that ARD is indeed active when sampling from the priors, in the sense that different dimensions have different lengthscales which are independently drawn from the prior. It would certainly be an interesting topic of future work for a network to identify the dimensions associated with the different kernel structures. However this would require some tweaking of our current architecture, due to its dimension-invariant nature.
>
> And thank you for bringing the interesting paper on stochastic grammar to our attention, we'll certainly add a mention of this.

---

> > ### Author Response · Authors · 2021-09-04
> > **post rebuttal discussion**
> >
> > Dear reviewer,
> >
> > Could you confirm that you have read the rebuttal and let us know if our answers have addressed the points you raised?
> >
> > Reviewer KsnS updated their score to 7 after the rebuttal and we would greatly appreciate if you could weigh on last time reasons to accept vs reasons to reject it in light of the response we provided.
> >
> > Kind regards

---

> > > ### Comment · Reviewer_VZVF · 2021-09-06
> > > **Score adjusted**
> > >
> > > Thank you for the clarification in the rebuttal. I raised the score to 6. Please consider providing discussion for 1d data.

---

> > > > ### Author Response · Authors · 2021-09-06
> > > > **post rebuttal discussion**
> > > >
> > > > Thanks for getting back to us quickly and for upgrading your score.
> > > >
> > > > Your suggestion of illustrating the method on a 1D experiment is indeed something that will be included in the revised version of the paper.

---

### Official Review · Reviewer_KsnS · 2021-07-19

**Rating:** 7
**Confidence:** 4

**Summary:**

The paper proposes a transformer-based neural network that can recommend appropriate kernels for Gaussian process regression. In the first step, the encoder uses a permutation-invariant version of multi-head self-attention to generate a representation of an entire dataset. This is further used by the decoder to recursively select a sequence of primitive kernels, which are then additively combined. The entire model is trained on a synthetic dataset consisting of samples drawn from GP priors with different kernels. Finally, the method is evaluated on a selection of regression datasets from the UCI repository in terms of prediction fit and runtime.

**Limitations And Societal Impact:**

Yes

**Main Review:**

The paper provides valuable and---to the best of my knowledge---novel contributions to the field of Gaussian process kernel selection. Compared to the closely related work by Liu et al., KITT can also recommend non-stationary kernels. Compared to search-based methods, the computational overhead during test time is significantly reduced. The paper is generally well-written and easy to understand; however, the structure of the presentation can be improved. Please see the detailed comments below.

The key strength of the paper is that the model is trained entirely on synthetic data while still being able to select good kernels for previously unseen, real-world datasets. In my opinion, the authors should emphasize this fact much more, as it is never really mentioned explicitly in the paper.

Its main weakness is that the description of the method lacks vital details (for example, about the data generation process) or spreads them across several sections, making it hard to understand the procedure. Furthermore, the evaluation seems slightly unfinished and relatively weak compared to the literature. Please see the comments below for more details.

Overall, I do not see any significant issues with the paper, and I deem that the idea is interesting and useful. For now, I would still rate it slightly below the acceptance threshold due to the issues mentioned in the comments. However, I am confident that the authors can address my comments/questions during the discussion phase, and I would be happy to raise my score if that is the case.


# Questions/Detailed Comments
* What training data was used for training the KITT network? That information is fragmented into several sections of the paper, the supplementary material and I even had to take a look at the code and related work (Liu et al.) to be completely sure. For example, the data generation process is only mentioned briefly in the paragraph about GP hyperparameter priors, the “training” paragraph states the dimensionality of the data, and the supplementary material contains the information that 200,000 samples are used in total. Why not compile this in a single paragraph about the training procedure? That would make the paper much easier to read and understand.
* It seems a bit odd to me that the AHGP model performs consistently worse than the simple RBF-ARD baseline, because, as you mentioned, the RBF kernel should be included in their formulation of the spectral mixture kernel. Were you able to reproduce results from the AHGP paper?
* How exactly does the “encoder-only” model mentioned in Fig. 4 work? I did not find this information anywhere in the paper. Is it perhaps the classification model from the ground truth recovery experiment and does it simply select a single primitive kernel expression from the vocabulary or a combination of the top-3 based on softmax values?
* Based on the experimental results, it seems that the encoder-only model’s performance is on par with the full model including the decoder. Is there any specific reason why you would ever want to use the decoder if the encoder-only model works just as well?
* The “Scalability” section (l. 232 ff.) feels a bit out of place. Its claims would need to be justified or you should at least provide references. In my opinion, the entire KITT pipeline also includes an $O(N^3)$ term as you optimize hyperparameters after the transformer suggests a suitable kernel, using the marginal likelihood gradient. I would suggest scrapping this section entirely and focusing on a more detailed empirical comparison of wall clock runtime instead, which is probably more relevant than asymptotic complexity for people who want to use your method anyway.
* Evaluation in the AHGP paper includes experiments in the domain of Bayesian optimization and Bayesian quadrature as well. It would be interesting to see how your methods performs for those tasks, but it is not strictly necessary in my opinion. However, your main selling point over the search-based methods is computational efficiency, so you should certainly include runtime comparisons for the UCI experiments or at least summarize the results and refer to the supplementary material. To make the comparison fair, I think that the KITT’s runtime should also include the optimization of hyperparameters.
* What is the effect of increasing/decreasing the maximal sequence length that the encoder is allowed to output? What happens if you allow products of more than two kernels. I only found the sentence “We use products of two primitive kernels as part of the base vocabulary and found that incorporating higher order products do not significantly change performance.” (l. 130 f.), but the supplementary material does not contain experimental results that support this statement.

## Minor:
* l. 57: “This quantity is available in closed-form for data modelled by Gaussian likelihoods.” You my want to explicitly state that “this” refers to the log marginal likelihood and not its global maximum to avoid confusion.
* Please cite published conference/journal papers instead of the arxiv version. E.g., Vaswani et. al was accepted at NIPS’17

## Typos:
* l. 122: “multiplying noise kernel” -> “multiplying a noise kernel”
* p. 5, fig. 2 caption: “heterescedastic” -> “heteroscedastic”
* l. 191: There is a footnote mark for the Set Attention Block, but the actual footnote does not seem to exist.


**Time Spent Reviewing:**

5

---

> ### Author Response · Authors · 2021-08-10
> **Reply to Reviewer KsnS**
>
> The reviewer raises a fair point by noting that a unified section to explain KITT's training procedure would be helpful. Furthermore we will modify Figure 1 to illustrate the concept that, for the purposes of training, the features are generated via random samples drawn from the labels.
>
> As mentioned in the general response above, we believe AHGP underperforms an RBF primarily because it is challenging to optimise the spectral mixture kernel, and as a result it often performs poorly in higher dimensional tasks (and performance tends to degrade further as more spectral components are added).  Nonetheless, we believe this should not detract from the excellent advancement that AHGP represents, as a first demonstration of how to generate viable hyperparameters in a fraction of a second.
>
> The 'encoder-only' model is, as you correctly guessed, the classification model specified on line 271. We shall clarify the text to make this clearer. The results from Figure 4 are top-3 based on the softmax values. As correctly noted, the current decoder doesn't offer a significant performance advantage over the encoder. However we the decoder is a valuable addition as it brings added flexibility, by opening up the potential for introducing more complex grammatical structures in the future.
>
> We have added a reference to the scalability of transformers. We prefer to leave this summary intact as we believe it is likely to be of interest to people working with datasets which are much larger than the UCI tasks under consideration.
>
> An important contribution of this work is a pre-trained model for GP kernel selection at scale. Many of the methods in the literature that deal with kernel selection do not decouple the kernel selection and hyperparameter training, in fact the kernel selection/search procedure involves fitting hyperparameters for multiple candidate models. This precludes their application at scale. Our approach is very distinct to greedy kernel selection procedures as it completely decouples the kernel selection step which relies on pre-training and downstream hyperparameter training. It is important to note that once pre-trained to a sufficient capacity, the model can be used at scale (several tasks) to infer kernels for high-dimensional regression. So while we agree it's important to highlight that for practical purposes, extra time will be taken to perform parameter inference, we also believe it is important to highlight the dramatic gain in speed for the specific task of kernel selection.
>
> For the case of triple (or higher) products, the different kernels become more challenging to distinguish (since the network only sees a small number of datapoints for each sample), so classification accuracy falls. The network then assigns low probability to each of these more numerous, more complex kernels. The net result is there isn't a statistically significant gain in experimental performance for this added complexity.  We don't believe this is a fundamental limitation however - to successfully train with more complex products would require training with significantly larger samples, which would in turn increase the computational complexity associated with training KITT.
>
> Thank you also for highlighting a few minor errors and incomplete citations which we shall fix.

---

> > ### Comment · Reviewer_KsnS · 2021-09-03
> > **Post-rebuttal**
> >
> > Thank you for taking the time to answer my questions! I feel that my concerns regarding the structure and the lack of clarity in some parts of the paper have been addressed. As mentioned in the main review I did not see any major issues with the paper to begin with, so I will update my rating to reflect this.

---

> > > ### Author Response · Authors · 2021-09-03
> > > **post-rebuttal**
> > >
> > > Thank you very much for taking the time to get back to us.
> > >
> > > We are glad that you decided to weigh again the pros and cons of the paper, and that you concluded that it was a good paper that should be accepted.

---

### Official Review · Reviewer_NPVS · 2021-07-29

**Rating:** 5
**Confidence:** 4

**Summary:**

The authors propose KITT, a novel method exploiting a transformer-based architecture to generate kernel recommendations. KITT has two main features: (1) agnostic to the length and dimensionality of the data; (2) it offers double permutation invariance. They demonstrate the effect of KITT on synthetic data compared with other baselines.

**Limitations And Societal Impact:**

The authors don't provide limitations of their work in the paper.

I mainly have the following concerns about the paper.
1. In my view, KITT apply an ensemble of tricks or technique together. Thus, an ablation study is important to illustrate the effect of different parts. For example, authors use a lot of places to describe heteroscedastic noise. I wonder what if we remove the noise and keep other parts the same.

2. As said in the Main Review part, all experiments are done on synthetic data, which is a simpler case in reality. Since Transformer is popular for NLP and CV tasks, can the authors show KITT on some small real-world data?


**Main Review:**

Originality: The kernel search problem is new for transformer-based architecture but it's not novel for kernel design. The authors just provide some prior kernel selections and select the result from them. I think one interesting part is Proposition 1 & 2. However, the main result is based on Zaheer et al. [33] and their Proposition can be seen as a direct extension.

Quality: I think the claim is intuitive and heuristic. For example, after defining a set of prior kernels, the authors apply addition and multiplication operations to generate more kernel candidates. The advantage is that the double permutation invariance is supported by their proof.

Clarity: In my view, the paper is not well organized and difficult to follow. For example, the contribution part in lines 36-47 is too scattered. Also, I think Section 3 is disordered: they use bold titles for each unparallel part. For instance, Encoder and Decoder should be the sub-part of Model Architecture.

Significance: The authors verify their KITT synthetic data and show its improvement compared with baselines. I am not sure about the improvement on the real-world data (like NLP dataset).

**Time Spent Reviewing:**

3

---

> ### Author Response · Authors · 2021-08-10
> **Reply to Reviewer NPVS**
>
> The reviewer's comments helped us realise that we ought to clarify which type of applications can and should be tackled with KITT. Although our approach is based upon a transformer architecture, its primary goal is to choose better kernels for GP models by learning to associate inherent structure in (X,y) datasets with suitable kernels. It is ultimately these GPs which are used for modelling tasks. As such, the domain of KITT is limited to applications for which Gaussian processes are relevant, and isn't particularly suited to NLP or CV tasks.
>
> We further clarify that the architecture we propose for this domain is significantly extended from the canonical NLP transformers; for instance, we incorporate invariance to sequence ordering and equivariance to feature ordering within each training set. We also omit positional encoding which was a key element for the NLP transformer.
>
> We certainly agree that it would be very concerning if our experiments were only based upon synthetic data. However, we can confirm that the UCI experiments displayed in Figure 4 consist of eight real world datasets. This includes data relating to the house prices in Boston and the hydrodynamic performance of yachts.  We will add an explicit statement that this reflects real world data.
>
> We certainly agree that ablation studies are important. As mentioned in the general response above, we shall modify the text to highlight some of the experimental results in the supplementary which help disentangle the contributions from different components. In particular, we note that on the UCI tasks, the kernels selected by KITT offer statistically significant performance improvements over randomly selected kernels. This is really the key result of the paper - that a suitably constructed transformer can assist in the selection of Gaussian Process kernels, and in a way which is dramatically faster than conventional methods.
>
> In light of the above comments, particularly the issue concerning real world data, we hope the reviewer will consider revising their score.

---

> > ### Author Response · Authors · 2021-09-04
> > **post rebuttal response**
> >
> > Dear reviewer,
> >
> > Could you confirm that you have read the rebuttal and let us know if our answers have addressed the points you raised?
> >
> > Reviewer KsnS updated their score to 7 after the rebuttal and we would greatly appreciate if you could re-assess one last time reasons to accept the paper vs reasons to reject it in light of the answers we provided to your remarks.
> >
> > Kind regards

---

> > > ### Comment · Reviewer_NPVS · 2021-09-07
> > > **post-rebuttal**
> > >
> > > Sorry for the late reply.
> > > As I said in the review, I think the paper lacks novelty since they apply the previous proposition directly.
> > > In the authors' rebuttal, they don't provide other comments about my novelty concern even though they claim that their approach is based upon Transformer architecture. Thus, I decide to keep my original score.

---

> > > > ### Author Response · Authors · 2021-09-08
> > > > **Clarifying the aspect of novelty**
> > > >
> > > > Many thanks again for taking the time to review this work, and apologies for not directly addressing your concern of novelty.
> > > >
> > > > To the best of our knowledge this is the first example of applying transformers (or indeed any neural network) to perform the task of kernel selection. We believe novelty is one of the key strengths of this paper, rather than a weakness. Reviewer KsnS noted that the paper makes "novel contributions", while VZVF adds the paper "presents a new approach".
> > > >
> > > > Zaheer et al provided a general framework to design network architectures which are permutation invariant. Insights from Zaheer have been used in several works to design task specific architectures. Our network architecture is designed to handle datasets as input which can be interpreted as set-structured data but must account for two invariances rather than one. In order to account for this, we use a composition of two attention-based encoders (Fig. 6 of supplementary). The two propositions aim to show how this specific compositional architecture preserves the invariances we desire. Although our propositions are architecture specific, they reveal how compositionality can be used to design other networks which need double permutation invariance.
> > > >
> > > > Kernel selection is a key problem in any GP modelling task. Given that we achieve better or similar performance compared to the SOTA in a fraction of the time (Figure 3 and 4), we foresee that the method we propose--or the future work that will be built upon it--has the potential to become the new best practice in kernel selection.

---

### Author Response · Authors · 2021-08-10
**Reply to reviews**

We would like to thank the three reviewers for their time and effort in reviewing the manuscript.

A common concern held by some reviewers relates to the structure of the presentation.  This led to some misunderstandings on the nature of the network, how it was trained, and the nature of the key experiments. We hope the responses below adequately address these concerns.

Another common question was why AHGP was outperformed by an RBF. While the AHGP result is of course not the focus of this work, it is possible to independently verify our finding by comparing to other published works who apply the RBF to UCI datasets. The spectral mixture kernel is challenging to generalise to higher input dimensions, and its successful application in the literature has largely been limited to one or two-dimensional problems (Wilson & Adams 2013; Wilson et al 2013).

The importance of an ablation study was also emphasised by two reviewers. We believe Table 4 in the Appendix provides an invaluable insight in this regard. It demonstrates that the kernel selected by KITT comfortably outperforms a random selection of kernels from the vocabulary. We hope this alleviates some of the concern that KITT's high performance is genuinely originating from the transformer. Further evidence of this can be seen in Figure 5 of the main text. The probability of correctly identifying the three highest performing kernels by chance is less than 0.003%.

---

### Decision · Program_Chairs · 2021-09-27

**Decision:**

Accept (Poster)

**Comment:**

The paper presents a new approach to select kernel functions via transformers. The proposed idea is interesting, and is a novel contribution to the field of Gaussian process and kernel selection with significantly reduced computational overhead during test time. Experimental results can be further strengthened by adding in more ablation studies. Moreover, organization and presentation can be improved.